# Super-Resolution Imaging of Plant Receptor-Like Kinases Uncovers Their Colocalization and Coordination with Nanometer Resolution

**DOI:** 10.3390/membranes13020142

**Published:** 2023-01-21

**Authors:** Jeremiah Traeger, Dehong Hu, Mengran Yang, Gary Stacey, Galya Orr

**Affiliations:** 1Environmental Molecular Sciences Laboratory, Pacific Northwest National Laboratory, Richland, WA 99354, USA; 2Division of Plant Sciences, University of Missouri, Columbia, MO 65211, USA

**Keywords:** fluorescence microscopy, flagellin-sensitive receptors, single-molecule, chitin-elicitor receptor kinase, immune response, phytology

## Abstract

Plant cell signaling often relies on the cellular organization of receptor-like kinases (RLKs) within membrane nanodomains to enhance signaling specificity and efficiency. Thus, nanometer-scale quantitative analysis of spatial organizations of RLKs could provide new understanding of mechanisms underlying plant responses to environmental stress. Here, we used stochastic optical reconstruction fluorescence microscopy (STORM) to quantify the colocalization of the flagellin-sensitive-2 (FLS2) receptor and the nanodomain marker, remorin, within *Arabidopsis thaliana* root hair cells. We found that recovery of FLS2 and remorin in the plasma membrane, following ligand-induced internalization by bacterial-flagellin-peptide (flg22), reached ~85% of their original membrane density after ~90 min. The pairs colocalized at the membrane at greater frequencies, compared with simulated randomly distributed pairs, except for directly after recovery, suggesting initial uncoordinated recovery followed by remorin and FLS2 pairing in the membrane. The purinergic receptor, P2K1, colocalized with remorin at similar frequencies as FLS2, while FLS2 and P2K1 colocalization occurred at significantly lower frequencies, suggesting that these RLKs mostly occupy distinct nanodomains. The chitin elicitor receptor, CERK1, colocalized with FLS2 and remorin at much lower frequencies, suggesting little coordination between CERK1 and FLS2. These findings emphasize STORM’s capacity to observe distinct nanodomains and degrees of coordination between plant cell receptors, and their respective immune pathways.

## 1. Introduction

Receptor-like kinases (RLKs) in the plant cell membrane provide highly efficient detection and response systems, which allow the cell to recognize adverse molecular patterns that may be present [1,2]. They may alert the cell to pathogens within the environment via the detection of pathogen-associated molecular patterns (PAMPs) or, additionally, detect signals indicating external cell damage and stress, such as extracellular ATP and other damage-associated molecular patterns (DAMPs) [3,4,5]. Detection of these patterns results in further downstream signaling, triggering the plant’s immune response.

RLKs and other membrane proteins have been observed localizing into small higher-order submicron-scale structures known as nanodomains, often observed to exist on a length scale of 100 nm, but occasionally as large as 400–500 nm [6,7,8,9]. In some plants, nanodomains were shown to facilitate interactions between related receptor complexes, leading to stable and effective functioning assemblies of proteins [10,11,12]. Remorin is a common plant membrane protein that has been shown to cluster in distinct nanoscale regions in the plasma membrane [13], often showing elevated expression in detergent-insoluble regions [14]. Because of these properties, it is well established as a marker protein for nanodomains. It has been observed to participate in the recruitment of various receptors into these localized domains [12,15,16]. By coordinating proteins with similar immune response pathways, these domains ostensibly enhance the spatiotemporal efficiency of interactions within proteins and receptors in similar response pathways [17,18].

Our prior work showed that activation of three different RLKs led to similar immune responses involving plant-triggered immunity [19]. The Flagellin Sensitive 2 (FLS2) receptor responds to flg22, a 22-sequence fragment that appears within bacterial flagellin, thus detecting bacterial pathogens at the plant extracellular space in both leaves and root hairs [20,21]. The P2K1 (DORN1) receptor responds to extracellular ATP, which is involved in the immune response to damage within the tissue [22]. CERK1 is a receptor for fungal chitin, providing further defense against fungal pathogens [23,24]. Stimulation of plant tissues with their cognate ligands results in an immune response leading to elevated Ca^2+^, reactive oxygen species (ROS), and mitogen-activated protein kinases (MAPKs) [25,26,27,28,29,30,31,32,33]. We also observed these receptors interacting with certain protein S-acyltransferases (PATs) [19,31], indicating that S-acylation of these RLKs by PATs results in specific localization of the receptors in nanodomains, as seen in animal cells [34]. Other work found that suppression of flg22 responses similarly affects chitin-mediated responses in a coordinated manner [20,35].

To our knowledge, there is little work showing direct interactions between FLS2, CERK1, and P2K1, despite these RLKs relying on similar membrane association via S-acylation and inducing similar downstream signaling effects. Zhou et al. demonstrated that FLS2 and BAK1 domains performed a similar role in chitin sensing as CERK1 [36]. Likewise, in addition to chitin and fungal resistance, CERK1 appears to perform an important role in bacterial resistance in a similar manner to FLS2 [32]. Despite these shared functionalities and components, it is unclear if these RLKs utilize the same signaling components within shared nanodomains or whether they merely contain similar downstream architecture, as prior work has shown that even clusters with similar components may be localized to distinct domains [15].

Prior work involving plant cell imaging observed the colocalization of receptors with remorin-containing nanodomains, as well as the colocalization of different RLKs to infer common functions and pathways. For example, Bücherl et al. used confocal microscopy to observe clusters of the receptors FLS2-GFP and BRI1-GFP within *Arabidopsis thaliana* leaf epidermal cells. They showed that their fluorescent signals colocalized with mRFP-Remorin nanodomains [15], indicating that both receptors colocalize with remorin but within distinct domains. Similarly, Liang et al. observed GFP-SYMREM and FLOT4-mCherry within *M. trunculata* root hairs, finding that scaffolding of nanodomains by FLOT4 is necessary before recruitment by remorin [10]. The dynamic construction of membrane domains may be a mechanism for coordinating the internalization of various receptors by enhancing the signal at certain receptor locations, such as the endocytic pathway for BRI1 [17] or by OMV-perceiving receptors in remorin nanodomains [37]. Mbengue et al. observed via confocal microscopy that FLS2 participated in similar endocytic routes as other pattern recognition receptors (PRRs) such as EFR and PEPR1 [38], exemplifying shared pathways for various PRRs. Other confocal studies inferring similar plant protein functions observed the colocalization of BIK1 and ARA6 [39] as well as the colocalization of BZR1 and ULP1a in *Arabidopsis* [40]. While these studies were effective at looking at broad correlations on an ensemble level, confocal microscopy’s resolution is restrained by the diffraction limit, providing constraints on observing true interactions between individual receptors below ~100 nm where RLKs may interact within nanodomains. Techniques outside of confocal microscopy have been employed to investigate RLKs, such as variable-angle TIRF microscopy [15,17], but these involve observing dynamic behavior, which results in a cost to spatial certainty.

To overcome these spatial limitations to techniques such as confocal microscopy and provide quantitative analyses of colocalizations within nanodomains at the relevant resolution below the diffraction limit, we employed stochastic optical reconstruction fluorescence microscopy (STORM) to colocalize receptors within fixed plant tissue at length scales starting below 100 nm. STORM relies on photoswitching fluorescent molecules that primarily remain in the “off” state over a long series of excitation frames, but are detected and localized when they stochastically appear in the “on” state. These locations are subsequently clustered and identified as individual fluorescent objects. This allows for reconstruction of the location of each molecule within a 20–30 nm spatial resolution, below the pixel size and diffraction limit [41,42]. STORM allows us to determine the proximity of various RLKs with the nanodomain marker protein, remorin, to determine their spatial coordination and infer their interactions within intact *Arabidopsis thaliana* root hair cells.

Here, we imaged N-terminal epitope-tagged FLS2, P2K1, and CERK1 receptors, as well as YFP-Remorin, within intact root hairs in transgenically modified *Arabidopsis thaliana* plants. N-terminal epitope tags are unique to our study and may prove advantageous in future work observing live tissue, as these plants will not have to be fixed to be labeled with fluorescent antibodies. All of the N-terminally tagged receptors were shown to be functional via complementation studies of each of the respective mutant lines (e.g., tagged FLS2 complemented the *fls2* mutant, etc.). Using super-resolution STORM imaging, we determined the locations and colocalization of these proteins with nm resolution. We used root hairs as an ideal tissue for detecting single molecules, required for STORM, due to their lack of pigments such as chlorophyll that interfere with localization of individual fluorescent molecules. We imaged root hairs stimulated by flg22 over time and quantified the internalization of FLS2 and remorin, observing long-term recycling of nanodomains containing both proteins. We also paired both remorin and FLS2 with two other RLKs, P2K1 and CERK1, to characterize their respective localization in nanodomains. We observed that FLS2 and P2K1 colocalized with remorin to a similar degree, although less colocalization was observed between the two receptors, suggesting their presence in distinct nanodomains. We also observed that CERK1 colocalized with FLS2 and remorin to a significantly lesser degree, suggesting little spatial coordination between CERK1 and FLS2. These observations highlight the involvement of remorin-populated nanodomains in a wide range of functions, where individual nanodomains may be populated by distinct receptors and devoted to distinct functions. These observations highlight the capacity for STORM as a tool for inferring coordinated behavior within the plant cell membrane. Overall, this work demonstrates the strength of STORM for further nanoscale investigation into plant immune system behavior.

## 2. Methods

### 2.1. Materials

Nuclease-free water, phosphate-buffered saline (PBS), and bovine serum albumin (BSA) were purchased from Fisher Scientific (Sumner, WA, USA). Ethanol, MES sodium salt, sucrose, glucose, glucose oxidase, catalase, mercaptoethanol, mercaptoethylamine tris-HCL, NaCl, driselase, MES, sodium zzide, Tween 20, and Triton X-100 were purchased from Sigma-Aldrich (St. Louis, MI, USA). Paraformaldehyde (PFA) was purchased from Electron Microscope Sciences (Hatfield, PA, USA). The Murashige and Skoog mixture was purchased from Caisson (Smithfield, United States). HA-Tag rabbit mAb AlexaFluor 555 and DYKDDDDK-Tag rabbit mAb AlexaFluor 488 (rabbit anti-FLAG) were purchased from Cell Signaling Technology (Beverly, MA, USA). GFP polyclonal antibody Alexa Fluor 488 was purchased from Invitrogen (Waltham, MA, USA). YFP-Remorin1.3 seeds were received from the Thomas Ott lab (Martinsried, Germany) [43].

### 2.2. Construction of Transgenic Plants

The HA or FLAG sequence-fused FLS2, P2K1, and CERK1 genomic fragments were generated by a two-fragment polymerase chain reaction (PCR) approach, in which primers including the HA or FLAG sequences were used in two separate PCR reactions. The primers used are listed in Appendix A. The positions of signal peptide were predicted by the SignalP 5.0 server [44]. Each fragment of FLS2, P2K1, orCERK1 was amplified by PCR from the genomic DNA of wild-type (Col-0) plants using gene-specific primers containing the HA or FLAG sequences. Overlapping PCR reactions were then performed to generate the whole FLS2, P2K1, or CERK1 genomic fragments with the HA or FLAG sequences inserted behind the signal peptide sequences of the receptors. The PCR products were cloned into a pDONR-Zeo vector (Invitrogen) by BP reaction and subsequently cloned into the binary vector pGWB1 (Invitrogen) by LR reaction.

Each construct was then transformed into the knockout mutant plants of the corresponding receptor using the *Agrobacterium tumefaciens* strain GV3101 by the “floral-dip” method [45]. The knockout mutants, including *p2k1* (SalK_042209), *fls2* (SALK_026801C), and *cerk1* (GABI-KAT 096F09) were obtained from the Arabidopsis Biological Resource Center (ABRC, Ohio State University, Columbus, OH, USA). The transgenic plants were selected by germination on 35 μg/mL hygromycin-containing half strength Murashige and Skoog (1/2 MS) medium. Homozygous T3 generation plants were used for following experiments. The functionality of the HA- or FLAG-tagged FLS2, P2K1, and CERK1 receptors were confirmed by their ability to complement the phenotypes of their knockout mutants. The double transgenic *Arabidopsis thaliana* plants, including YFP-Remorin1.3 × HA-FLS2, YFP-Remorin1.3 × HA-P2K1, YFP-Remorin1.3 × HA-CERK1, FLAG-P2K1 × HA-FLS2, and FLAG-FLS2 × HA-CERK1 were generated by crossing two transgenic lines, and the F1 generation plants were used for imaging.

### 2.3. Arabidopsis Growth

Seeds were sterilized first by rinsing twice in 70% ethanol, 30% Milli-Q water, and 0.1% Triton X-100, followed by rinsing twice in ethanol. After drying, seeds were incubated at 4 °C for 3 days in nuclease-free water. Seeds were pipetted into an agar dish comprising 1% sucrose, 0.22% Murashige and Skoog mixture, and 0.05% MES. These plates were placed in an incubator with 16-h light/8-h dark growth conditions at ~22 °C and 50% humidity.

After 14 days of growth, seedlings were removed from plates and fixed by placing them in 4% PFA under vacuum conditions with subsequent agitation after ~5 min to remove oxygen bubbles. For seedlings that were stimulated, plants were first placed in 200 nM flg22 for a fixed amount of time before aspirating the flg22 solution and replacing it with 4% PFA. Seedlings remained under vacuum for 1 h and were then removed, where they were then rinsed and stored with PBS at 4 °C.

### 2.4. Immunostaining

Cell walls were first digested with 0.2% driselase in 2 mM MES at pH ≈ 5.7 at 37 °C for 15 min. Following this, the plasma membrane was permeabilized by 1% Triton X-100 in PBS at room temperature for 20 min. Before staining, seedlings were placed in a blocking buffer of 1% BSA, 0.1% between 20, and 2 mM sodium azide. Antibody staining was prepared in this solution, where rabbit anti-HA was prepared at 1:200 dilution, rabbit anti-FLAG was prepared at 1:50 dilution, and anti-GFP was prepared at 1:200 dilution. Anti-GFP antibodies were used to stain YFP-Remorin, anti-HA antibodies were used for HA-FLS2 and HA-CERK1, and anti-FLAG antibodies were used to stain FLAG-FLS2 and FLAG-P2K1. Plants were incubated overnight in staining solution at 4 °C. Plants were subsequently rinsed thoroughly three times, consisting of aspirating 8–10 times with room-temperature PBS and waiting 10 min between aspirating to ensure thorough washing, before storing in PBS at 4 °C until imaging.

### 2.5. Fluorescence Microscopy

Imaging buffer, consisting of 50 mM Tris, 10 mM NaCl, 10% glucose, 56 μg/mL glucose oxidase, 34 μg/mL catalase, 1% 2-mercaptoethanol, and 1% mercaptoethylamine, was prepared for imaging. Root hairs were removed from seedlings and placed on a glass bottomed dish in imaging buffer under a glass coverslip. The dish was mounted on a 100× oil immersion objective on an Olympus IX-71 microscope. The root hair was illuminated with a 542 solid-state laser (Crystalaser), where the collected light passed through a 572 nm filter with a 90% transmission width of 28 nm (for imaging Alexa 555 antibodies) or, alternatively, a 488 nm solid-state laser (Spectra-Physics) where the collected light passed through a 520 nm filter with a 90% transmission width of 28 nm (for imaging Alexa 488 antibodies). Both lasers were set to emit at 5 mW/cm^2^. Images were collected with an iXon Ultra 897 Andor Electron Multiplying CCD camera. Upon illumination, 512 × 512-pixel images at 160 nm per pixel were captured for 8190 frames at 10 fps and stored as 16-bit FITS files.

### 2.6. Image Processing

Image processing is described in detail in previous work, using in-house MATLAB scripts that are available upon request [46,47]. Centroids for blinking fluorescent objects were identified and used to generate a blinking density map, where the density of the map was constructed to indicate the number of blinking events within 20 nm (the error of fluorescence localization within cells). These blinking events were grouped into clusters, indicating an individual fluorophore. The membranes were then outlined so that the points within the outline area were designated as objects on the membrane. The outlined region was selected based on the membrane regions that remained in the same focal plane or “in-focus”. The locations of fluorescent objects in the different channels were aligned with an iterative closest point (ICP) algorithm for further analysis to account for shifts between channels.

Linear densities of membrane proteins were measured by taking observed protein counts within our outlined region along the cell membrane and normalizing these counts by half the perimeter of the outline (assuming that the cross-membrane dimension width is negligible compared to the length along the membrane). Simulated membrane protein density was measured by generating a line for each of these lengths and randomly populating these lengths with the same quantities of proteins as in the experiment and repeating this 100 times. Random simulation was chosen over experiments with a control pair known to not colocalize, as non-colocalizing pairs of membrane proteins would have been anticorrelated due to colocalizing in separate domains rather than non-correlated. To account for ICP alignment artificially enhancing colocalization within our experiments, we also aligned simulated clusters via ICP.

## 3. Results

### 3.1. Quantitative Analysis of FLS2 and Remorin Colocalization in the Membrane at Increasing Time Points following Ligand Stimulation

To quantify the cell membrane colocalization of FLS2 and the marker protein remorin in response to ligand stimulation, we stimulated live *Arabidopsis thaliana* with 200 nM flg22, chosen as a relatively high concentration of flg22 shown to reliably induce an immune response [48,49]. This was followed by fixation with 4% PFA at increasing time points post-stimulation. After immunostaining HA-FLS2 and YFP-Remorin with anti-HA-AlexaFluor 555 and anti-GFP-AlexaFluor 488, respectively, we observed remorin and FLS2 via widefield microscopy with single molecule detection sensitivity and subsequent analysis via STORM. Images of root hairs with localized FLS2 and remorin at high resolution are shown in Figure 1. Red circles indicate colocalization of FLS2 and remorin within 100 nm (Appendix A), a value chosen as a commonly-referenced nanodomain size [6] The areas analyzed only include receptors along the cell membrane. Additional super-resolution images are shown in Appendix A.

These images indicate a wide variation between root hairs in the number of membrane proteins identified using super-resolution localization. On average, upon elicitation with flg22, the number of remorin and FLS2 in the membrane decreased, resulting in a lower frequency of their colocalization. After ~30 min post-stimulation, the number of FLS2 and remorin appear to increase again, suggesting recycling or restoration of FLS2 and remorin to the membrane.

To further quantify the number of remorin and FLS2 localized to the root hair membrane, we measured the linear density of these membrane proteins along the membrane. This linear density was defined as the number of proteins per unit length along the observed root hair membrane. Remorin and FLS2 appear to populate the membrane at ~0.55–0.6 μm^−1^ within root hairs that had been unstimulated by flg22 (Figure 2a). These values drop to a minimum of ~0.25 µm^−1^ after 10 min post-stimulation. The linear density of these proteins is increased back to ~0.55 µm^−1^ at around 30 min post-stimulation, roughly 85% of the initial density. In order to highlight that significant internalization was due to stimulation, we included a secondary control sample of plants unstimulated by flg22 in the brief period before fixation. These controls were shown to be statistically similar to measurements at time 0 and at 90 min post-stimulation, but statistically distinct from measurements 15 min post-stimulation (*p* < 0.05, 2-sided *t*-test). Images are shown in Appendix A.

Similar to the linear density of membrane proteins, the linear density of FLS2 and remorin colocalization within 100 nm also undergoes a drop followed by a recovery (Figure 2b). With no stimulation, colocalization occurs at ~0.17 μm^−1^. Around 15 min post-stimulation, the linear density of colocalization drops to ~0.04 μm^−1^, eventually recovering to ~0.11 µm^−1^ at 90 min. The alternate unstimulated control showed a linear density measurement significantly similar to the measurement at t = 0 but statistically distinct from the linear density of colocalization 15 min after stimulation (*p* < 0.005, 2-sided *t*-test). Unlike the density of receptors, the linear density of this control was significantly different from the measurement at 90 min (*p* < 0.05), highlighting that total post-stimulation recovery has not been observed at this time.

Simulations of randomly populated membranes with FLS2 and remorin, using the same areas and quantities as their experimental analogs, were conducted and repeated 100 times to determine whether the changes in colocalization of the two proteins over time, following ligand stimulation, occur in a coordinated fashion. While the linear density of remorin/FLS2 colocalization in our experiments was generally within the range of 0.03–0.2 µm^−1^, colocalization in simulations was generally lower and within the range of 0.03–0.09 µm^−1^ (Appendix A). The experimental linear density of colocalization at 0 min was significantly higher than the random population, indicating that there is coordination between FLS2 and remorin. However, 15 min post-stimulation, the linear density of colocalization between experiment and simulation no longer held a significant difference. This difference appeared to diverge again after 90 min post-stimulation. The coordination between FLS2 and remorin after 15 min was effectively random, implying that the uptake of remorin and FLS2 involves coordinated pairs on shared nanodomains, leaving behind uncoordinated membrane proteins.

To better understand the relationships between each FLS2 receptor and its nearest remorin, we quantified the distance between each FLS2 and its nearest-neighbor remorin in a continuous fashion. The cumulative distribution functions (CDFs) are plotted for comparison in Figure 3. Before stimulation at time 0, over 50% of nearest-neighbor distances are below 400 nm. The distribution of nearest-neighbor distances appears to increase at longer stimulation times of ~15–30 min, before decreasing again at 90 min. The distributions at t = 0 min vs. t = 90 min below 500 nm fail the null hypothesis of being samples of the same continuous distribution to a high degree of significance (*p* < 0.001, two-sample Kolmogorov–Smirnov test). Thus, despite a period of recovery time, repopulation is incomplete by 90 min.

We also included the CDFs of nearest-neighbor distances of simulated FLS2 remorin pairs randomly distributed along the root hair membrane. Notably, the distributions of experimentally measured nearest-neighbor distances are lower than the simulated distributions, except for at 30 min, where the CDFs follow a similar trajectory. When performing a two-sided Kolmogorov–Smirnov test for the experimental and simulated CDFs below 500 nm at 30 min, the two distributions fail to reject the null hypothesis, suggesting statistical similarity. Comparisons at other time points reject statistical similarity between experimental observation and random distribution, indicating that we only observed random distribution of FLS2 and remorin at 30 min and that, after that time point, FLS2 and remorin become more coordinated along the membrane. This further suggests that, while internalization of remorin and FLS2 is coordinated, recovery back to the membrane is not coordinated. The pairs become significantly more coordinated at 90 min post-stimulation, suggesting pairing occurs in the membrane after recovery.

### 3.2. Quantitative Analysis of P2K1 and CERK1 Colocalization with FLS2 or Remorin

To determine whether other RLKs show similar spatial behaviors to FLS2, we quantified the colocalization of P2K1 and CERK1 with remorin, as well as the colocalization of these two receptors with FLS2. The following double-labeled transgenic *Arabidopsis thaliana* plants were used: YFP-Remorin × HA-P2K1, YFP-Remorin × HA-CERK1, FLAG-P2K1 × HA-FLS2, and FLAG-FLS2 × HA-CERK1 plants. The plants were fixed and labeled with their respective antibodies as follows: anti-GFP-Alexa488, anti-FLAG-Alexa488, or anti-HA-Alexa555. We used STORM to localize and compare FLS2’s locations with those of the P2K1 and CERK1 (Figure 4 and Appendix A). P2K1 and CERK1 appeared to colocalize with FLS2 to a lesser extent compared with remorin and, in a few cases, there was no colocalization within 100 nm (Figure 4f). P2K1 was more prone to occupy separate domains than FLS2, sometimes within regions several micrometers in size (Figure 4c,d). CERK1 was prone to populating even larger regions that did not contain FLS2 (Figure 4e,f).

Despite having a similar density of receptors along the root hair membrane (Appendix A), the degree of colocalization varied between the different receptor pairs (Figure 5). Colocalization, distinguished as localization of FLS2 within 100 nm of another protein, was measured as 0.18 ± 0.07 µm^−1^ for the remorin/FLS2 pair. P2K1/FLS2 had a lower density of colocalization at roughly 0.1 ± 0.06 µm^−1^. FLS2/CERK1 had a much lower linear density of colocalization, 0.04 ± 0.02 µm^−1^, which was significantly lower than the remorin/FLS2 pair (*p* < 0.01, two-sided *t*-test). Remorin colocalized with P2K1 at a similar rate to FLS2 (0.15 ± 0.05 µm^−1^), while remorin colocalized with CERK1 at a significantly lower frequency of 0.08 ± 0.03 µm^−1^.

To further characterize the distances between receptor pairs and between the receptors and remorin, we plotted the distances between each FLS2 and its nearest neighbor in the opposite channel, as well as the nearest-neighbor distances between the other receptors and remorin (Figure 6). Notably, the CDF for the remorin/FLS2 pairs was statistically similar to the CDF for the remorin/P2K1 pairs (*p* > 0.5), suggesting that the presence of remorin is coordinated with the presence of P2K1 and FLS2 in a similar manner. All other CDFs for other pairs were significantly lower than these two pairs, suggesting that coordination between these protein pairs was not as frequent. Both remorin/FLS2 and remorin/P2K1 CDFs show that FLS2 and P2K1 occupy nanodomains with remorin. However, the distances between FLS2 and P2K1 tend to be longer than their respective distances from remorin, suggesting that the nanodomains these two RLKs occupy are distinct.

In order to compare linear densities of colocalization more directly while accounting for varying expression levels of proteins within root hairs, we normalized the colocalization linear density by the linear density of both membrane proteins in the root hairs. These normalized linear densities are more similar to each other as a result of increased expression levels leading to increased rate of colocalization, with normalized colocalization densities of ~0.35–0.75 µm. (Appendix A). Simulations were performed on the same length along the membrane as those used experimentally to compare colocalizations and distances observed via experiment to randomly populated receptors and remorin. All simulated normalized colocalization linear densities had similar values of around ~0.27–0.3 µm. Because of this similar measurement and the fact that these were all generated randomly, this indicates that this normalization is apt for comparing root hairs with different expression levels. Notably, all experimental normalized densities were significantly different from the randomly generated densities (*p* < 0.05), except for the FLS2/CERK1 pair, highlighting their lack of colocalization.

Simulations were performed on the same length along the membrane as those used experimentally, to compare distances observed via experiment to randomly populated receptors and remorin. In all cases, the distributions of experimentally observed nearest-neighbor distances were smaller than randomly deposited simulated protein pairs at length scales relevant to micro- and nanodomains (less than 1 µm). Notably, the CDFs appear to differ greatly from simulation, even at low nearest-neighbor distances of below 100 nm (Appendix A). When performing a two-sided Kolmogorov–Smirnov test between below 100 nm, the remorin/FLS2 and remorin/P2K1 distributions of experimentally observed distances are significantly different from simulated randomly distributed pairs (*p* < 0.05). This highlights that coordination between remorin and FLS2 or P2K1 is observable within nanodomain-scale length scales. This contrasts with the other pairs, where differences between experimental and simulated CDFs are not statistically distinct.

The size of membrane nanodomains has been a subject of debate. Measurements generally indicate nanodomain sizes of several hundreds of nanometers [6] but they have been observed as large as 500 nm [43]. To gain insight to the size of the nanodomains, we compared the experimental nearest-neighbor CDFs with the CDFs of the random simulation. We identified the point of inflection where the CDF of the experiment began to approach the simulation, i.e., the nearest neighbor distance that is maximally different between the experiment and simulated CDF (Figure 7).

As remorin is the marker protein for nanodomains, we identified the maximum difference between experiment and simulation in the nearest-neighbor distance between the receptors and remorin (Figure 7a–c). We found the maximum difference to be between 300 and 400 nm (350 ± 40 nm). Notably, this maximum difference was higher (~440 nm) for the FLS2/P2K1 and FLS2/CERK1 pairs (Figure 7d,e), which presumably are pairs of receptors that commonly occupy distinct nanodomains. This may be an indicator of the characteristic length scale of the nanodomains in which the receptors tend to be coordinated with remorin.

## 4. Discussion

Using super-resolution fluorescence microscopy, we were able to localize individual membrane receptors in intact plant tissue and colocalize them within a 100 nm length scale. Nanodomains are estimated to have a size of between 100-300 nm [50,51,52]. By using STORM along with the nanodomain marker, remorin, we were able to localize the receptors within that length scale and, furthermore, estimate the size of these nanodomains to be ~300–400 nm.

Proteins appeared to occupy the length along the membrane with a linear density of ~0.4–0.6 µm^−1^ and appropriate comparisons with other literature requires converting to an area density. Our measured linear density values translate to a surface area density of ~0.2–0.4 µm^−2^. This is somewhat lower than the nanodomain density within *Arabidopsis thaliana* measured by Gronnnier et al. of 0.75–1.0 µm^−2^ [53], but similar to the density of remorin domains in *Medicago trucatula* root hairs reported by Liang et al. at ~0.35 µm^−2^ [10]. Notably, these measurements reveal a somewhat lower density of these receptors in root hairs compared to leaves. Bücherl et al. observed a FLS2 density of ~2.5 µm^−2^ in *Arabidopsis thaliana* leaves [15], while Cui et al. observed a much larger density value of ~12 µm^−2^ for FLS2 in leaves [54]. This may indicate the relative importance of immune response proteins within leaves compared to root hairs, and future work with these techniques may benefit from looking at leaf tissue instead.

Upon stimulation with flg22, plant cells elicit a response within the first 30 min [31,39,55,56], including internalization of FLS2, which is required for flagellin signaling [21]. We observed this effect upon stimulation with 200 nM flg22, where a maximum internalization of remorin and FLS2 occurred at around 15 min, in agreement with the previously reported timescale [56]. In addition to the membrane density loss of receptors and remorin, the CDF distributions of nearest-neighbor distance also decrease upon stimulation with flg22. Interestingly, we observed a recovery of these molecules in the membrane in the long term (around 90 min). The linear density of 100 nm colocalization at 90 min returned to ~85% of its value before stimulation, and the CDF at 90 min post-stimulation w statistically similar to the CDF at 5 min post-stimulation (*p* > 0.1). These observations give a rough recovery time and suggest that full recovery, indistinguishable from baseline, takes longer than 90 min. Smith et al. observed that reactive oxygen species (ROS) production in leaves after stimulation with flg22 reached peak activity levels within 10–15 min and returned to baseline levels after nearly 40 min [56], suggesting that post-elicitation recovery occurs on a longer timescale than flg22 reaction. Jeworutzki et al., observing a change in membrane potential depolarization in root hair cells following stimulation with 10 nM of flg22, and measured a recovery time of baseline membrane composition near to 70 min, closer to the timescale of our measurement [57].

Both the linear density of colocalization and the CDFs are statistically distinct from a random distribution of the receptors and remorin up to 15 min post-stimulation, where their distribution becomes similar to a random distribution. This observation suggests that the internalization of FLS2 and remorin is coordinated and that contents of entire nanodomains are internalized as a whole in response to flg22 stimulation. This interpretation agrees with prior work looking at responses to flg22, where both FLS2 and remorin became enriched in detergent-resistant membranes upon ligand stimulation [58] and disruption of clustering of FLS2 into these detergent-rich domains inhibited FLS2 endocytosis, as well as immune signaling, through this pathway [59].

Interestingly, while we observed the recovery of proteins after 30 min post-stimulation, the linear density of colocalization was similar to that of random distribution, and at 30 min the experimental CDF was similar to what would occur for a random distribution of proteins. After 90 min, however, remorin and FLS2 became significantly more coordinated than randomly distributed membrane proteins. These observations suggest that the mechanisms for the return of proteins to the plasma membrane and the re-colocalization or coordination of FLS2 and remorin in the recovery process occur in several steps. Figure 8 illustrates a proposed sequence of events, where vesicles containing FLS2 nanodomains are taken up upon flg22 stimulation as a whole (step 2), a phenomenon which has been observed before between FLS2 and BRI1 [38]. Between internalization and recovery, some remorin and FLS2 molecules become uncoordinated (step 3), and return to the membrane in an uncorrelated fashion (step 4). Prior work indicates that FLS2 undergoes sorting into specific laminated endosomes and early endosomes, depending on the pathway for elicitation, and separation could occur during this process [60]. After long-term recovery past 90 min, the membrane proteins that have returned to the membrane become colocalized or coordinated (step 5), although notably, even at 90 min post-stimulation, we observed a smaller degree of colocalization between FLS2 and remorin, implying total recovery is longer than this time scale.

Remorin is considered to be a marker protein for various types of nanodomain [43,61]. As FLS2 has been shown to colocalize with remorin within these nanodomains [15,62,63], the distributions and measurements of colocalization between FLS2 and remorin may be a good point of comparison for other colocalization pairs. Notably, we found that P2K1 colocalizes with remorin at a similar frequency as the remorin/FLS2 pair, but the colocalization of P2K1 with FLS2 occurs at a significantly lower frequency. Together, these observations suggest that P2K1 and FLS2 mostly occupy distinct remorin nanodomains. Interestingly, less than 40% of FLS2 appeared to colocalize within 200 nm of remorin, thus, much of the remorin-populated nanodomains may be devoted to other distinct sensing roles such as that of P2K1 [15]. CERK1 colocalizes significantly less with both FLS2 and remorin compared with P2K1 or FLS2 and remorin, emphasizing that the CERK1 pathway may be further distinguished from FLS2 and P2K1 sensing. In the limited literature comparing interactions between these RLKs, FLS2 and CERK1 appear to perform similar pathogenic sensing roles [32,36]. The lack of spatial overlap may suggest that redundancies in these roles may benefit from separate nanodomain architectures. Accounting for the varying expression levels of the receptors by normalization, however, indicates that all three RLKs colocalize with remorin significantly more often than random, emphasizing that remorin nanodomains likely play a significant role in their function. The poor colocalization of FLS2 with other RLKs below 200 nm suggests that the nanodomains containing these RLKs perform distinct roles within separate, uniquely acting clusters.

## 5. Conclusions

STORM was used to distinguish colocalization and coordination between membrane receptors and remorin, as well as between receptor pairs. Using this approach, we were able to discern colocalization patterns at sub-diffraction and nanodomain length scales. We were able to quantify the FLS2/remorin colocalization at nanoscale, shedding light on their coordinated internalization following ligand stimulations, and on the multi-step mechanism of their recovery back to the membrane. We identified quantitative differences between the colocalization of FLS2/remorin and P2K1/remorin or CERK1/remorin, indicating that these three receptors likely occupy distinct nanodomains. While we found some degree of spatial coordination between FLS2 and P2K1, little coordination was found between CERK1 and FLS2, indicating distinct pathways. Our detailed, nanometer-scale findings demonstrate that STORM provides unique capability to investigate the architecture and behavior of membrane receptors in intact plant tissue, shedding light on their coordination and the potential coordination between their immune pathways.

## Figures and Tables

**Figure 1 membranes-13-00142-f001:**
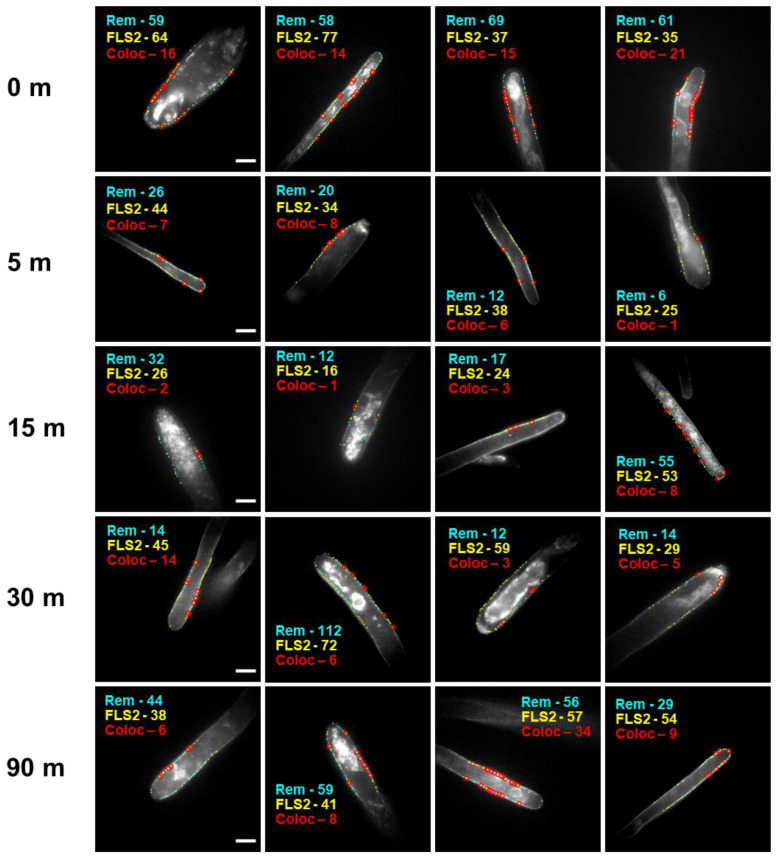
Representative super-resolution images of *A. thaliana* root hairs stimulated with 200 nM flg22 for a set length of time and then fixed, indicating localization of remorin (cyan), FLS2 (yellow), and colocalized remorin-FLS2 pairs within 100 nm (red circles). The scale bars indicate 10 µm. Note the steady drop in receptor density over time, which is largely restored within 90 min.

**Figure 2 membranes-13-00142-f002:**
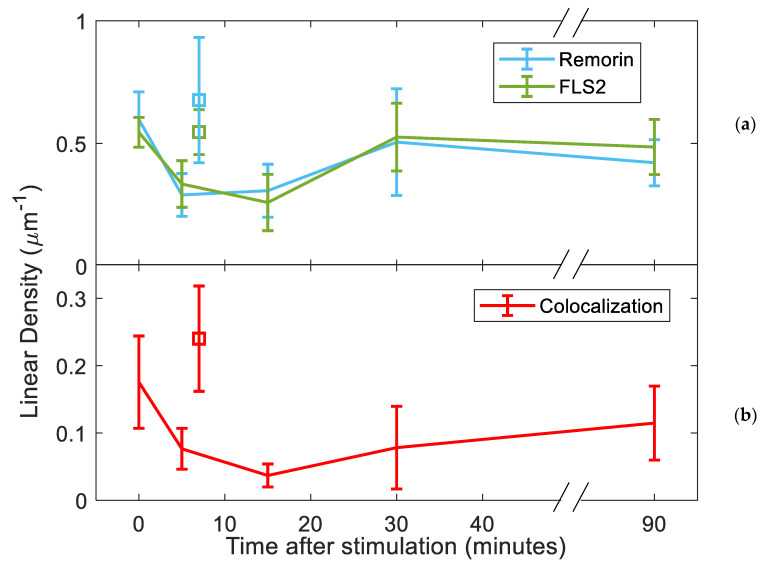
Time series post-stimulation with 200 nM of flg22 showing the linear density of (**a**) FLS2 and remorin, (**b**) colocalization of FLS2 within 100 nm of remorin. Squares indicate alternative sample of unstimulated points for comparison.

**Figure 3 membranes-13-00142-f003:**
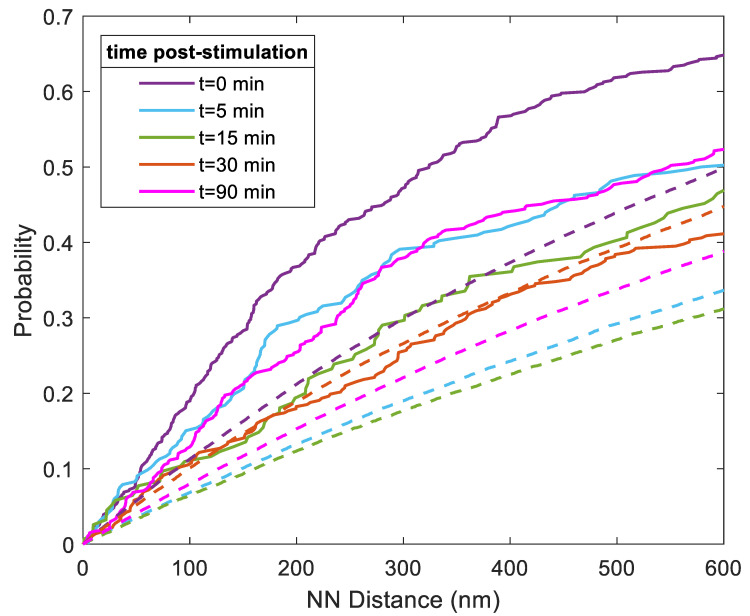
Cumulative distribution functions of the distance of each FLS2 to its nearest-neighbor remorin at increasing time points (see color key) post-ligand stimulation. Solid lines indicate experimentally observed values. Dashed lines indicate simulated random distribution values. Measurements for nearest-neighbor distances were performed up to 40,000 nm (Appendix A), while this figure details length scales relevant to nanodomains.

**Figure 4 membranes-13-00142-f004:**
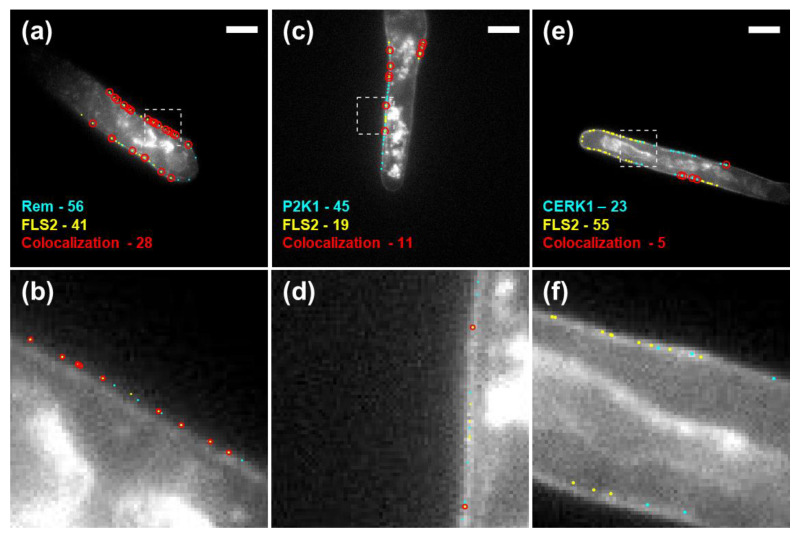
Example of super-resolution images of *Arabidopsis thaliana* root hairs, detailing images of remorin/FLS2 (**a**,**b**), P2K1/FLS2 (**c**,**d**), and FLS2/CERK1 (**e**,**f**). The areas marked by the squares in (**a**,**c**,**e**) are enlarged in (**b**,**d**,**f**), respectively. For (**b**,**d**), red circles are 100 nm radius, depicting the colocalization range. No colocalization within 100 nm is found in (**f**). Scale bars are 10 µm.

**Figure 5 membranes-13-00142-f005:**
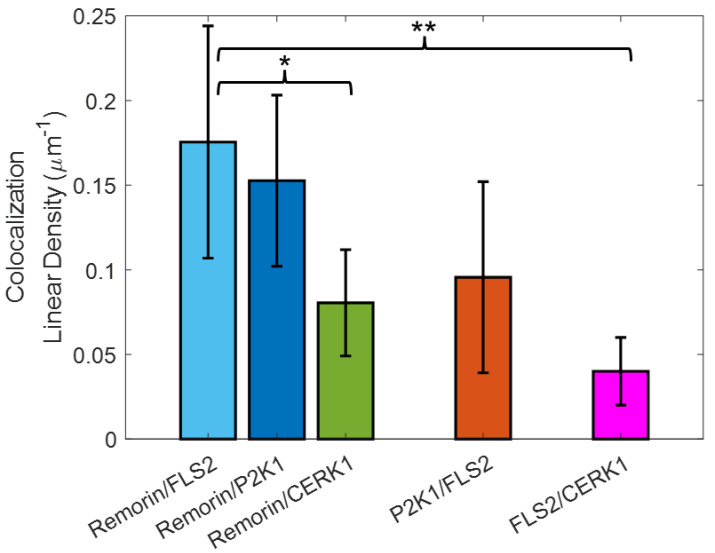
The linear density of colocalization within 100 nm of various pairs of membrane proteins along the root hair membrane. * *p* < 0.05, ** *p* < 0.01.

**Figure 6 membranes-13-00142-f006:**
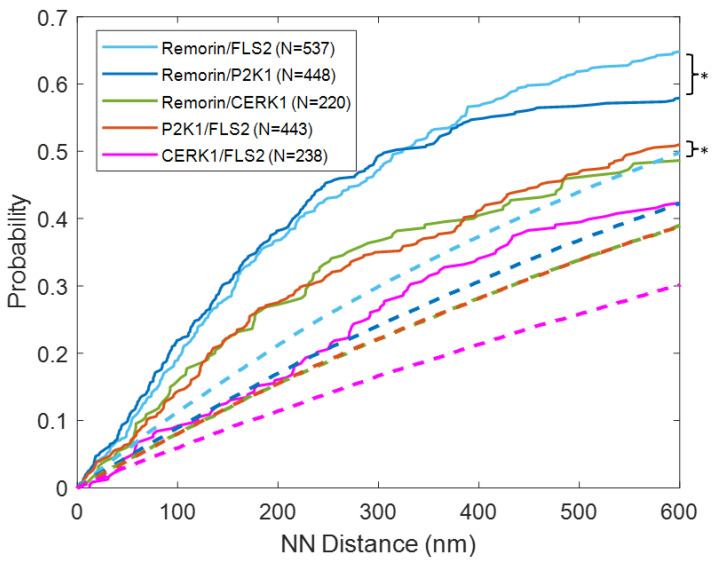
Cumulative distribution function distances from each identified FLS2 to its nearest-neighbor protein in the other channel, or from P2K1 or CERK1 to remorin. Dashed lines indicate simulated nearest-neighbor distances. Asterisk pairs indicate two-sided Kolmogorov–Smirnov significance similarity >0.05 for nearest-neighbor distance below 500 nm.

**Figure 7 membranes-13-00142-f007:**
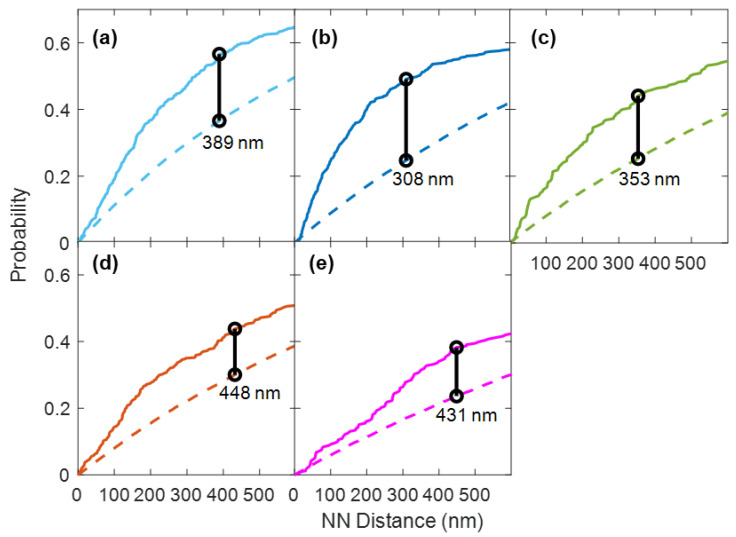
Cumulative distribution functions (CDFs) of the nearest-neighbor distance between (**a**) FLS2 and remorin, (**b**) P2K1 and remorin, (**c**) CERK1 and remorin, (**d**) FLS2 and P2K1, and (**e**) FLS2 and CERK1. Solid line indicates experimental CDFs. Dashed lines indicate simulated CDFs. Black lines indicate the x-values where the maximum difference between experiment and simulation is found.

**Figure 8 membranes-13-00142-f008:**
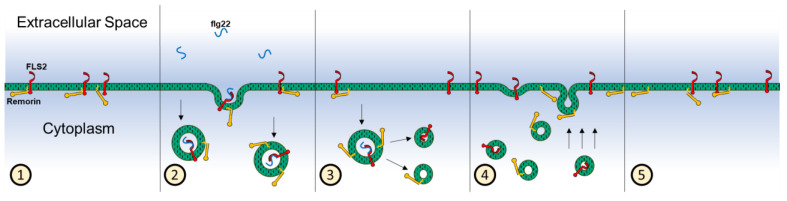
Illustration of the proposed model for protein internalization upon flg22 stimulation. (1) Native state of membrane proteins within the cellular membrane before stimulation, where FLS2 and remorin are typically colocalized in nanodomains. (2) Stimulation with flg22, where FLS2 binding with flg22 results in the internalization of nanodomains including remorin. (3) Smaller vesicles bud from the original endocytic vesicles, separating FLS2 and remorin into different vesicles. (4) Recovery of the proteins in the membrane, where remorin and FLS2 return to the membrane separately. (5) 90 min post-stimulation, where the density of the proteins in the membrane has recovered, and most remorin and FLS2 have colocalized or coordinated back into the same nanodomains.

## Data Availability

Publicly available datasets and images were analyzed in this study. This data can be found here: https://search.emsl.pnnl.gov/?project[0]=projects_51203 (accessed on 9 November 2022).

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
