# Peer review of "Super-Resolution Imaging of Plant Receptor-Like Kinases Uncovers Their Colocalization and Coordination with Nanometer Resolution"

_membranes, 2023, doi:10.3390/membranes13020142_

Round 1
Reviewer 1 Report
Brief summary;
This manuscript aims to understand a behavior of a plant immune receptor, FLS2, and a microdomain marker, remorin, during pathogen ligand stimulation. Using super-resolution imaging strategy, STORM, it succeeds to quantify single molecules of the two membrane proteins before and after stimulation and suggests that the two proteins are internalized together during ligand stimulation and separated in different compartments before recover to the membrane. It also measures the distance between remorin and three immune receptors and discusses degrees of coordination between the receptors and the microdomain.
General comments
1. Although I recognize the considerable efforts of making double reporter transgenic lines, immuno-stain, taking images, and analyzing images, I find some weakness of lacking negative controls. For the experiments of flg22 stimulation in FLS2/remorin plants, I would like to see water or antagonistic peptide treatment as negative control that are not decreasing density nor colocalization during the treatment. While I understand that FLS2 internalization by flg22 is well accepted phenomenon, the new approach using STORM should be demonstrated as a reproducible tool.
2. Although it is not feasible to make another double reporter transgenic line, it would have been better if there were another double reporter transgenic of two membrane proteins that are not colocalized at all as a negative control for measuring distance of several pairs of proteins and showing similar experimental values to simulation values.
3. Figure 5 is not so impressive as figure S4 is showing similar trend to me.
Specific comments
1. Line 124-130; show primer sequences. How are the positions of signal peptide determined?
2. Line 142; Cite the source of the YFP-remorin plant. Which one of remorin genes is used?
3. Line 186-201; cite which software is used.
4. Line 195; Is there any objective rules to define start and end points of membrane? This could be highly variable depending on how to define and could produce very different outcomes.
5. Line 216-217; Are there enlarged images of yellow signal only, blue signal only and the both colors with red circles so that I can believe those red circles as colocalization?
6. Line 248; it must be 0.03-0.09 not 0.03-0.009.
7. Line 293-295; Are these two sentences referring figure 4? It is not obvious to me.
8. Line 334-336; I’m not fully understanding how distance blow 100 nm can be measured in 160 nm per pixel images with less than 10 nm resolution.
Line 336-341; The difference between experimental and simulation in Remorin/CERK1 looks more than the difference in Remorin/FLS2 to me.
Reviewer 2 Report
The authors elegantly showed how to use Super-Resolution microscopy to learn the co-localization behavior of different RLKs and the nanodomain marker, and expanded the method to the Exploration of time-dependent ligand inducible internalization. It is very likely that by distance measurement between RLKs and remorin proteins with STORM, the authors were able to provide a solution to study the distances between different nanodomains. However, there are still a few things that need to be improved for the current manuscript.
1. Introduction:
Major:
- The relationship between nanodomain and RLKs internalization was missing in the introduction part;
- please add a few sentences about remorin proteins' function in the nanodomain, e.g why remorin protein can be used as a nanodomain marker?
Minor:
Line 55-57- Unclear.
How did the authors define “the same signaling pathway”? do you mean the RLKs’ downstream responsive components? Line 57, “distinct domains” is it meaning “nanodomains”? It is a bit confusing here based on the examples the author provided. From these examples, we could see different RLKs have function specificity but also share some common downstream regulators. Please clarify the authors’ opinion here.
2. Method
Major:
-Which member of remorin proteins was used in this study?
-Line155, why to use 200nN flg22? If the concentration of a ligand is higher than the RLK binding saturation, it can also trigger PM RLKs recycling.
-Which software/plug-in was used for image processing?
-What about the expression level of each protein? Does the expression level of protein affect its co-localization? Does protein expression level affect the ligand-induced signal restoration in the current study?
3. Result:
Minor:
Line235 Figure2,
Did the authors normalize the data before comparing different proteins' “linear density”?
Line258-259 about Figure3,
Why the cumulative probability at the y-axis was not reached 100% (1)?
Lin328-329 Figure6
How many data points were collected for performing the statistics (should add n=?)?
Line359
In previous publications or the authors' other biochemistry experiments, do FLS2 and P2L1 interact with each other? Also, do FLS2 and CERK1 interact with each other? It would be very appreciated if the authors could address this point either in the introduction part or the discussion part.
3 Discussion
Minor:
Lin 372, µm -2 should be µm -1
Round 2
Reviewer 1 Report
I think the authors addressed all the questions and comments from the reviewers and I am convinced that their claims are justified. I believe the manuscript meets the journal's standard and shall be accepted.